# Is NRXN1 Gene Expression an Important Marker of Treatment of Depressive Disorders? A Pilot Study

**DOI:** 10.3390/jpm11070637

**Published:** 2021-07-06

**Authors:** Aleksandra Skiba, Monika Talarowska, Janusz Szemraj, Piotr Gałecki

**Affiliations:** 1Department of Adult Psychiatry, Medical University of Lodz, 91-229 Lodz, Poland; piotr.galecki@umed.lodz.pl; 2Department of Clinical Psychology, Institute of Psychology University of Lodz, 91-433 Lodz, Poland; talarowskamonika@wp.pl; 3Department of Medical Biochemistry, Medical University of Lodz, 92-215 Lodz, Poland; janusz.szemraj@umed.lodz.pl

**Keywords:** depression, *NRXN1* gene, expression

## Abstract

**Aim:** Due to the fact that *NRXN1* is associated with neurodevelopmental disorders, the aim of this study was to investigate the role of the *NRXN1* gene in the etiology and epigenetics of depression by comparison of *NRXN1* mRNA expression and *NRXN1* protein level expression in patients suffering from depression versus healthy controls, as well as to search for clinical variables related to expression of the analyzed gene. **Material and Methods:** A total of 180 people aged 19–64 qualified for the study. The experimental group consisted of 97 people who were psychiatrically hospitalized, diagnosed with recurrent depressive disorders (F33) or who met the diagnostic criteria of a depressive episode (F32) according to ICD-10. The control group included 83 healthy people who volunteered to participate in the study. A sample of peripheral blood was obtained from people who were positively qualified to participate in the study—twice in the experimental group and once in the control group for genetic testing. Sociodemographic variables and data on the course of the disorder were also gathered. Patients were examined on study entry and at the end of the hospitalization with the Hamilton Depression Scale. Obtained data were analyzed statistically. The study was approved by the University’s Bioethics Committee. **Results:** The gene expression of *NRXN1* at both mRNA and protein level significantly differs and it is lower in the experimental group compared to expression in healthy people. The difference in gene expression of *NRXN1* at both the mRNA and protein levels between the first and second measurement in the experimental group is also significant. The result demonstrates a higher expression level in the first measurement and lower expression level in the second measurement when reported depression symptoms are less severe. **Conclusions:** Results concerning expression of *NRXN1* may play an important role in further researches about the etiopathogenesis of depressive disorders such as looking for depression biomarkers and identifying evidence which may be relevant to personalize treatment for depression.

## 1. Introduction

Depressive disorders are one of the leading causes of psychiatric hospitalization and one of the leading causes of disability worldwide. Forecasts point to a steady increase in the number of patients, especially in developing countries. Depressive disorders are a serious health problem for patients that often leads to disorganization in their professional, family, and social lives. Depression often appears as a coexisting symptom in other diseases. Conducted clinical trials show that a significant group of people with an initial diagnosis of a depressive episode will have a recurrent course of the disease in the future [1]. About 20% of patients diagnosed with recurrent depressive disorders (RDD) experience two phases of remission during their lifetime while about 60% of them experience three or more [2].

Currently, it is assumed that the conditions of this disorder are multifactorial: biological, psychological, and social. The biopsychosocial model which includes biomedical and psychosocial approaches, emphasizes their interrelationships and interdependence [3]. The mechanism of inheritance in the etiology of depressive disorders is not sufficiently explained. Family researches, including twin studies, indicate the role of heredity in depressive disorders. The results of the study of monozygotic twins show that the incidence of recurrent depressive disorders ranges from 43% to 53% [4]. The complex model of inheritance assumes the interaction of several genes. The studies show that the regions of chromosomes 15q, 17p, and 8p [5,6] as well as 12q23.3-q24.11 and 13q31.1-q31.3 may contain genes associated with depressive disorders [7]. The monoamine hypothesis emphasizes the nature of abnormalities in neurotransmission of serotonin (5-HT), noradrenaline (NA), dopamine (DA), and acetylcholine (ACh) [8,9]. Structural changes in the brain of patients with depressive disorders are characterized by a reduction in volume in the frontal lobes, orbital prefrontal cortex, anterior cingulate cortex, hippocampus, and amygdala [10,11]. These regions are directly associated with the symptoms of RDD [12]. In studies on the kynurenine pathway, a decreased level of kynurenic acid was found in the RDD group [13]. The inflammatory theory of depression connects adverse effects of stress with a greater likelihood of depressive symptoms with the influence of hormonal, biochemical, and epigenetic factors. In the RDD group, increased parameters of the active inflammatory process are observed—mainly interleukins 1 and 6, TNF-α, and interferon gamma [14,15]. A theory that combines the biological, psychological, and environmental factors is the neurodevelopmental theory of depression [16]. 

Neurexins are presynaptic adhesive proteins that play a role in connecting neurons at the synapse. They consist of three genes: *NRXN1*, *NRXN2*, and *NRXN3*. They are mostly located on the presynaptic membrane and contain a single transmembrane domain [17]. They nucleate the trans-synaptic signaling network that assists the properties of synapses and are susceptible to impairments in neuropsychiatric disorders. Mutations in genes encoding neurexins are associated with Tourette syndrome, autism, and schizophrenia [18]. Other research shows the relationship between neurexins and impulsive behavior as well as alcohol dependence [19]. Neurexin 1 (*NRXN1*) is a protein responsible for synaptic homeostasis, development of glutamatergic and GABAergic synapses, balance of stimulation, and inhibition of synaptic formation and functioning in the central nervous system [20]. Research suggests that *NRXN1* is associated with neurodevelopmental disorders characterized by abnormalities in synaptic transmission, such as autism and intellectual disability. Levels of *NRXN1* expression in the prefrontal cortex are changed in schizophrenia and bipolar disorder in comparison to healthy people. Studies suggest that *NRXN1* expression is the highest in the human prefrontal cortex during critical developmental moments. It is associated with the onset and diagnosis of a range of neurodevelopmental disorders and may be an important variable in neuropsychiatric disorders [21]. Another studies confirm the importance of exon deletion near the 5′ end of *NRXN1* in the expression of neurodevelopmental disorders [22]. For a child with Pitt-Hopkins syndrome, the heterozygosity for two mutations in the *NRXN1* gene was identified [23]. Studies in groups of patients diagnosed with autism spectrum disorder confirm the *NRXN1* mutations in sick sibling pairs [24,25]. Loss of heterozygosity for *NRXN1* and *NRXN2* in mice leads to phenotypes associated with autism and schizophrenia [26]. A 380 kb deletion of *NRXN1* occurred in a woman with Asperger’s syndrome, anxiety, and depression as well as in four of her children diagnosed with autism, anxiety disorders, developmental delay, and speech delay. This change was not observed in her healthy child [27]. Research indicates that peripheral blood determinations which are expressed at the mRNA level and at the protein level for genes largely reflect the expression in the central nervous system [28]. 

Due to this fact, the aim of the study was to verify the expression of gene *NRXN1* at the mRNA and protein level in patients suffering from depression versus healthy controls, as well as to search for clinical variables related to expression of the analyzed gene. The purpose of the study was also to investigate the role of the *NRXN1* gene in the etiology and epigenetics of depression. It may help to make the correct diagnosis, look for depression biomarkers and to identify evidence which may be relevant to personalize treatment for depression such as targeted therapies, especially in treatment-resistant depression.

## 2. Material

A total of 180 people aged 19–64 were qualified and took part in the study (mean: M = 38.25; standard deviation: SD = 10.01; median: Me = 35), included 119 women (66%, aged 19–64; M = 38.27; SD = 12.25; Me = 36) and 61 men (34%, aged 19–62; M = 38.21; SD = 14.50; Me = 32). 

The experimental group (DD) consisted of 97 people who were psychiatrically hospitalized, diagnosed with recurrent depressive disorders (F33) or who met the diagnostic criteria of depressive episode (F32) according to ICD10. The experimental group included people aged 19–64 (M = 43.22; SD = 12.78; Me = 46), included 64 women (66%, aged 19–64; M = 42.23; SD = 12.05; Me = 45) and 33 men (34%, aged 19– 62; M = 45.12; SD = 14.09; Me = 51).

The control group (CG) consisted of 83 healthy people who volunteered to participate in the study. The control group included people aged 20–60 (M = 32.45; SD = 10.73; Me = 28), included 55 women (66%, aged 20–60; M = 33.65; SD = 10.86; Me = 29) and 28 men (34%, aged 20–58; M = 30.07; SD = 10.25; Me = 27). 

Table 1 consist of basic information about people who took part in the study.

Patients suffering from depression who were additionally diagnosed with other mental disorders, addictions, traumatic central nervous system injuries, severe neurological and chronic inflammatory diseases were excluded from the study. The same rules of exclusion were also applied to the control group. Data on the illness course (duration of the disease, number of hospitalizations, and number of episodes) were obtained using the Composite International Diagnostic Interview (CIDI). In addition, information about sociodemographic variables (age, sex, education, and place of residence) was collected. Severity of depression at inclusion to the study and at the end of hospitalization was evaluated using the Hamilton Depression Rating Scale (HDRS) for DD. A sample of peripheral blood was obtained from people who were positively qualified to participate in the study—twice in the experimental group (at the beginning and end of hospitalization) and once in the control group for genetic testing. The aim of double blood sampling in the experimental group was to verify if the expression of the studied gene may change when the improvement in mental state was expected and reported depression symptoms were less severe. It was also related to the end of treatment in the hospital. All of the patients were treated with typical antidepressant treatment, mostly with SSRIs (selective serotonin reuptake inhibitor). All obtained data were analyzed statistically. The study was approved by the University’s Bioethics Committee. All study participants were familiarized with the information about the study, have given an informed consent to participate in the research. They could withdraw it at any stage of the study without any consequences, including further treatment in the experimental group.

## 3. Methods

Total RNA isolation from the patients’ peripheral blood mononuclear cells was performed with the use of InviTrap Spin Universal RNA Kit (Stratec molecular, Berlin, Germany) in accordance with the manufacturer’s recommendations. Absorbance of isolated RNA was measured using a spectrophotometer (Picodrop™ Microliter Spectrophotometer VWR International, LLC, Radnor, PA, USA) at λ = 260 nm in order to determine total RNA concentration. The quality of total RNA was checked with Agilent RNA 6000 Nano Kit (Agilent Technologies, Santa Clara, CA, USA) in accordance with the manufacturer’s recommendations. The real-time PCR reaction was conducted using TaqMan^®^ RNA Reverse Transcription Kit (Applied Biosystems, Waltham, MA, USA) in accordance with the manufacturer’s recommendations. To calculate relative expression of mRNA genes, the Ct comparative method was used [29].

Concentration of *NRXN1* protein in the serum of patients and controls was determined by ELISA testing, using the Human *NRXN1* Elisa kit (MyBiosource, San Diego, CA, USA) in accordance with the manufacturer’s recommendations. β-actin was used as endogenous control of protein concentrations in the samples and was labeled with Human Actin Beta Kit (ACTb) ELISA Kit (BMASSAY, HaiDian District, Beijing China) in accordance with the manufacturer’s recommendations. 

The 17-item Hamilton Depression Rating Scale (HDRS) was used to assess the severity of depressive disorder in the experimental group. The severity of symptoms was assessed on a four-point scale (from 0 to 4 points in 13 items and from 0 to 2 points for the rest) [30,31].

Obtained data were analyzed statistically using the STATISTICA PL, version 13.1 (TIBCO Software, Palo Alto, CA, USA). The statistically significant differences were considered at *p* < 0.05. Selected descriptive and statistical inference methods were used to compile the data. The arithmetic mean (M), median (Me), and standard deviation (SD) were calculated. The Shapiro–Wilk test was used to assess the normality of the variables’ distribution. *T*-test or Mann–Whitney U test were used to compare the gene expression at the mRNA and protein levels. A *T*-test was also used to compare the degree of depression in the RDD group between the measurements. The Spearman’s rank correlation coefficient and regression analysis methods were used to study relationship between variables. ANOVA was used to compare pairs of dependent and independent variables.

## 4. Results

A statistically significant difference was found in expression of the *NRXN1* gene at the mRNA level in the experimental group—DD and the control group—CG. Expression of the *NRXN1* gene at the mRNA level is significantly lower in the DD group than in the CG (*p* = 0.001). The difference in expression of the *NRXN1* gene at the protein level also turned out to be statistically significant. Expression of the *NRXN1* gene at the protein level is significantly lower in the DD group than in the CG (*p* = 0.001). The results of gene expression are described in Table 2 and on Figure 1 where all significant differences are presented.

To extend the previously presented results, the analysis of the relationship between expression of the studied gene on both levels in the experimental group was conducted. It shows a significant correlation between expression of the *NRXN1* gene at the protein level and a significant correlation between expression of the same gene at the mRNA level. This is a weak positive correlation. The greater expression of the *NRXN1* gene at the protein level, the greater expression of this gene at the mRNA level (Spearman’s rank correlation coefficient = 0.28; *p* < 0.05).

What is more, the analysis of the relationship between expression of the studied gene in the control group shows a significant correlation between expression of the *NRXN1* gene at the protein level and expression of the same gene at the mRNA level. This is a very strong positive correlation. The greater expression of the *NRXN1* gene at the protein level, the greater expression of this *NRXN1* gene at the mRNA level (Spearman’s rank correlation coefficient = 0.95; *p* < 0.05). 

The division of both research groups in terms of age revealed significant difference related to expression of the *NRXN1* gene at the protein level. Expression of the *NRXN1* gene is significantly higher in the younger group of respondents (*p* = 0.004). The metric variable “age” was divided into two groups according to the median. The results are described on the Figure 2.

There is no significant relationship between expression of the studied gene and sex. There is also no significant relationship between clinical variables and expression of the studied gene in DD group.

The division of DD group in terms of age revealed significant differences related to the severity of depression symptoms. The metric variable “age” was divided into two groups according to the median. In both measurements (number 1 *p* = 0.001; number 2 *p* = 0.020), the results of the HDRS test are higher in the older group of respondents—greater severity of symptoms. The results are described in Figure 3 and Figure 4.

The division of DD group in terms duration of the disease revealed significant difference related to the severity of depression symptoms. The metric variable “disease duration” was divided into two groups according to the median. The result of the HDRS test in the first measurement is higher in the group with a shorter disease duration (*p* = 0.004). The results are described on Figure 5.

There is no significant result of an influence of sociodemographic and clinical course variables on expression of *NRXN1* gene at both mRNA and protein levels in the DD group in the second measurement.

The results of the relationship between expression of *NRXN1* gene in the DD group from the first and second measurement revealed statistically significant, positive correlations. Expression of *NRXN1* in the first measurement at the mRNA level correlates with expression of *NRXN1* in the second measurement at the mRNA level and at the protein level. Both relationships are weak (Spearman’s rank correlation coefficient = 0.28 for both; *p* < 0.05).

Expression of *NRXN1* in the first measurement at the protein level correlates with expression of *NRXN1* in the second measurement at the mRNA level and at the protein level. Both relationships are very strong (Spearman’s rank correlation coefficient = 0.89 and 0.88; *p* < 0.05).

Paired difference tests between the measurements were also performed. The following relationships were found:*NRXN1* expression at the mRNA (RQ) level—second measurement > first measurement*NRXN1* expression at the protein level (ng/mL)—second measurement > first measurement

The results are described in Table 3.

In order to assess which factors significantly influenced expression level of the studied gene in the groups, a regression analysis was performed for expression of *NRXN1* at the protein level with two predictors—expression of *NRXN1* at the mRNA level and depression morbidity (group). A statistically significant result is obtained which indicated that the presence of depression and expression of the *NRXN1* gene at the mRNA level are significant predictors of expression of the *NRXN1* gene at the protein level (*p* = 0.001; standard error BETA = 0.058; confidence interval = 0.95). All the results are described in Table 4.

Significantly lower result in the DD group of the HDRS test in the second measurement was found when comparing to the results from the first measurement (*p* = 0.001). The result is described in Table 5 and in Figure 6.

## 5. Discussion

The scientific literature contains no specific research results and scientific reports about expression of the *NRXN1* gene in the group of patients diagnosed with depressive disorders. There are few results concerning other mental disorders—ASD, Pitt-Hopkins syndrome, schizophrenia, or intellectual disability [32]. Kirov et al. in their research prove that the deletions of *NRXN1* confer a substantial increase in the risk of schizophrenia [33]. The differences in the level of *NRXN1* in comparison with reported data from researches in patients with schizophrenia and bipolar disorder [34] may be due to the fact that depression is a common disease in the population and characterized by a heterogeneous etiology, i.e., there are also differences in the contribution of genetic factors of depression. Moreover, *NRXN1* and NLGN3 are differentially expressed in the cerebral cortex and hippocampus in study in mice, which may be responsible for the changes in synaptic plasticity during aging associated with neuroplasticity [35]. This may have an impact on the potential neurexin depletion in depression and its influence on the synaptic machinery in depression development. 

This study showed that the expressions of the *NRXN1* gene at both mRNA and protein levels in patients with recurrent depressive disorders significantly differ from the expressions of this gene in the health control group. Expression was lower in the group of patients with depressive disorders. However, another study shows the increases in *NRXN1-α* and β expression in bipolar disorder and schizophrenia [21]. The relationships of mutations and expression of the *NRXN1* gene published in the literature confirm its importance in neurodevelopmental and neuropsychiatric disorders [36,37] so this may be a significant indication of its importance in depressive disorders. Another important finding is the demonstration of significant relationships between the expression of the studied gene in the experimental group. The greater the expression of the *NRXN1* gene at the protein level, the greater the expression of this gene at the mRNA level. Interestingly, the same relationship was shown for the *NRXN1* gene in the control group. However, there is no possibility of comparison results for a selected gene in the group of patients with depressive disorders in the available scientific literature. Studies concerning the effect of *NRXN* polymorphisms and clozapine treatment in the group of patients with schizophrenia show its deletion [38,39]. Research by Lam et al. proves that *NRXN1-alpha* plays an important role for the efficient establishment of neural stem cells as well as in neuronal differentiation while differentiated cells from healthy control individuals and an individual with autism spectrum disorder carrying bi-allelic *NRXN1-alpha* deletion [40].

Then, significant relationships were demonstrated between *NRXN1* expression in the first and second measurement. The relationships turned out to be positive. Expression level was higher in the second measurement. Selected data collected from the respondents on sociodemographic variables also turned out to be significant with the expression of the studied gene in the experimental and control group. The differences in expression were statistically significant when the whole study group was divided by age. In the further part of the study, it was shown that the occurrence of depression and the expression of the *NRXN1* gene at the mRNA level were significant predictors of the expression of the *NRXN1* gene at the protein level. 

The last research aim was to compare the severity of depression in the experimental group between the measurements and to assess the impact of other sociodemographic variables. Significantly lower results of the HDRS test in the second measurement compared to the results from the first measurement were found. This may indicate an important improvement in the mental state in the experimental group. What is more, this influence may be linked to the conclusions regarding the differences between expression of *NRXN1* in both measurements in the experimental group. Furthermore, in both measurements, the results of the HDRS test were higher in the older group of respondents, which may indicate an increase in the severity of depressive symptoms with age, what is associated with an increasingly frequent diagnosis of depression in the elderly and decreased quality of life in this group [41]. The results of the HDRS test in the first measurement were higher in the group of respondents with a shorter disease duration.

## 6. Conclusions

Gene expression of *NRXN1* at both mRNA and protein levels may play an important role in the etiopathogenesis of depressive disorders and these expressions significantly differ from the expressions of these genes in healthy people. These conclusions can encourage research into further studies that are oriented to new methods of prevention, diagnosis, and treatment of depression.

## 7. Limitations

The most important limitation of this study is the significant age difference between the experimental and control group which could have influenced the analyses performed. It would be worth verifying the significant relationships in the next study with a larger and proportional research group as well as using respondent-driven sampling (RDS) as an example of statistical method. A significant limitation may also be the possibility of qualifying people with co-morbidity affecting the obtained results, although exclusion criteria were also applied upon inclusion. It should be added that the survey and interview were carried out to minimize this risk. It would also be worthwhile to verify whether the specific applied pharmacological treatment or the combination with psychotherapy may also affect the expression of the studied gene. 

## Figures and Tables

**Figure 1 jpm-11-00637-f001:**
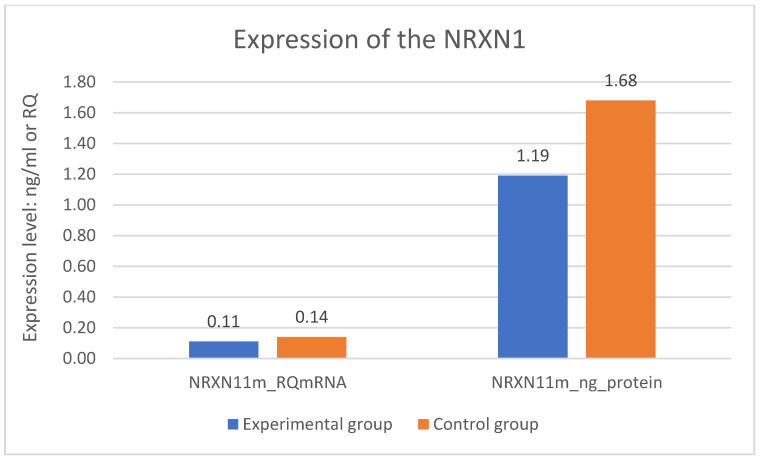
Expression of the NRXN1 gene in the examined groups.

**Figure 2 jpm-11-00637-f002:**
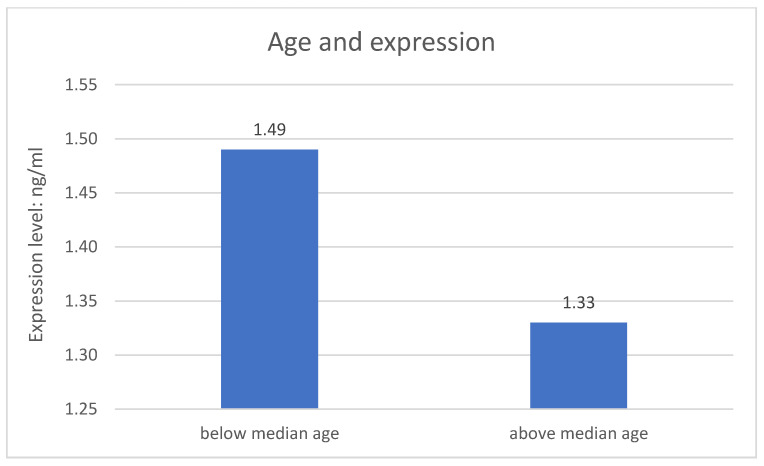
Age and expression of the studied gene.

**Figure 3 jpm-11-00637-f003:**
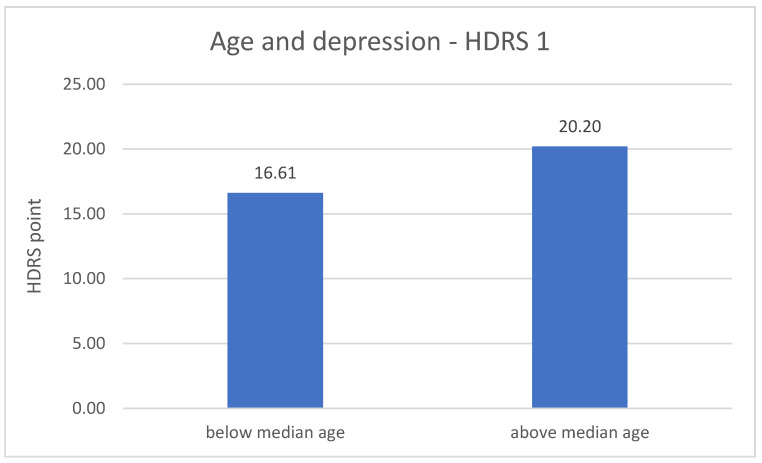
Age and the severity of depression—HDRS 1 measurement.

**Figure 4 jpm-11-00637-f004:**
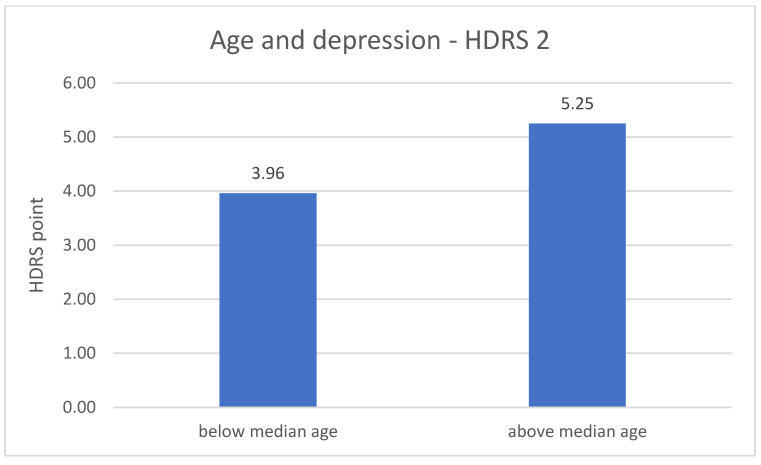
Age and the severity of depression—HDRS 2 measurement.

**Figure 5 jpm-11-00637-f005:**
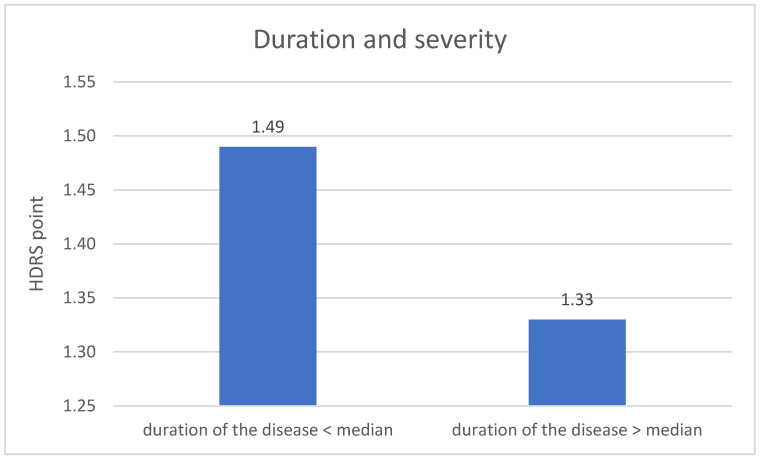
Duration of the disease and the severity of depression—1 measurement.

**Figure 6 jpm-11-00637-f006:**
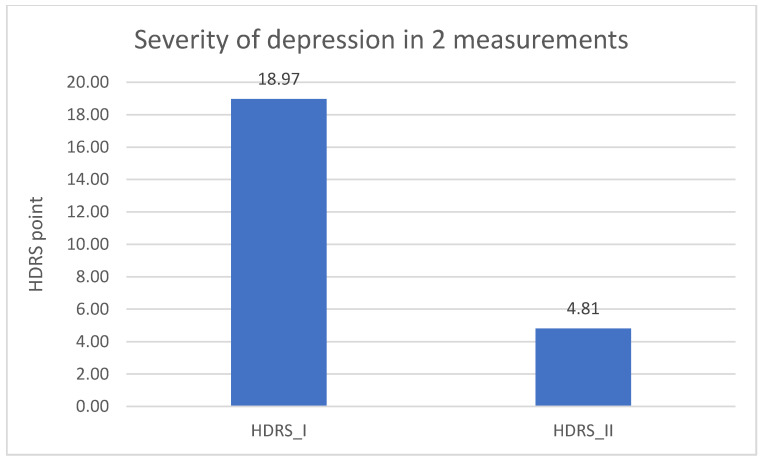
Severity of depression in the experimental group—1 and 2 measurement.

**Table 1 jpm-11-00637-t001:** Demographic characteristics: experimental and control group.

Variable	N	Age	Age M	Age SD	Age Me
Total group	180	19–64	38.25	10.01	35
Total group—women	119 (66%)	19–64	38.27	12.25	36
Total group—men	61 (34%)	19–62	38.21	14.50	32
Experimental group	97	19–64	43.22	12.75	46
Experimental group—women	64 (66%)	19–64	42.23	12.05	45
Experimental group—men	33 (34%)	19–62	45.12	14.09	51
Control group	83	20–60	33.65	10.86	29
Control group—women	55 (66%)	20–60	33.65	10.86	29
Control group—men	28 (34%)	20–58	30.07	10.25	27

N—number; M—mean; SD—standard deviation; Me—median.

**Table 2 jpm-11-00637-t002:** Expression of the NRXN1 gene in the examined groups.

Variable	M DD	M CG	*t*	*p*	N DD	N CG	SD DD	SD CG
NRXN11m_RQmRNA	0.11	0.14	−5.589	0.001 *	97	83	0.053	0.018
NRXN11m_ng_protein	1.19	1.68	−11.151	0.001 *	97	83	0.341	0.221

NRXN11m_RQmRNA—NRXN1 expression in 1 measurement mRNA level (RQ); NRXN11m_ng_protein—NRXN1 expression in 1 measurement protein level (ng/mL); DD—experimental group (group of patients with depressive disorders); CG—control group; N—sample; *p*—level of statistical significance (* statistically significant *p* < 0,05); SD—standard deviation; *t*—value of *t*-test; M—mean.

**Table 3 jpm-11-00637-t003:** Results of paired difference tests between the measurements of the expression of the studied gene in the experimental group.

Variable	M 1st Measurement	SD 1st Measurement	M 2st Measurement	SD 2st Measurement	*p*
NRXN11m_RQmRNA: NRXN12m_RQmRNA:	0.107	0.053	0.118	0.024	<0.05
NRXN11m_ng_protein: NRXN12m_ng_protein:	1.191	0.341	1.402	0.298	<0.05

NRXN11m_RQmRNA—NRXN1 expression in 1 measurement mRNA level (RQ); NRXN11m_ng_protein—NRXN1 expression in 1 measurement protein level (ng/mL); NRXN12m_RQmRNA—NRXN1 expression in 2 measurement mRNA level (RQ); NRXN12m_ng_protein—NRXN1 expression in 2 measurement protein level (ng/mL); SD—standard deviation; M—mean.

**Table 4 jpm-11-00637-t004:** Variable regression summary—expression of the studied gene in the examined groups including predictors: expression of the studied gene at the mRNA level and the occurrence of depression.

Variable	Predictor	R	R^2^	SR^2^	F	*p*
NRXN11m_ng_protein	Depression	0.70	0.49	0.49	86.43	0.001 *
	NRXN11m_RQmRNA					

NRXN11m_RQmRNA—NRXN1 expression in 1 measurement mRNA level (RQ); NRXN11m_ng_protein—NRXN1 expression in 1 measurement protein level (ng/mL); F—F statistic value; *p*—level of statistical significance (* statistically significant *p* < 0,05); R—value of the regression coefficient; R^2^—squared regression coefficient; SR^2^—standardized R^2^.

**Table 5 jpm-11-00637-t005:** Severity of depression in the experimental group—1 and 2 measurement.

Variable	HDRS_I M DD	HDRS_I SD DD	HDRS_II M DD	HDRS_II SD DD	F	*p*
HDRS_I: HDRS_II	18.97	6.63	4.81	3.93	270.248	0.001 *

HDRS_I—Hamilton Depression Rating Scale 1 measurement; HDRS_II—Hamilton Depression Rating Scale 2 measurement; DD—experimental group (group of patients with depressive disorders); SD—standard deviation; M—mean; F—F statistic value; *p*—level of statistical significance (* statistically significant *p* < 0,05).

## Data Availability

The data analyzed in the study are available upon request to the authors of the article.

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
