# Peer review of "Is NRXN1 Gene Expression an Important Marker of Treatment of Depressive Disorders? A Pilot Study"

_jpm, 2021, doi:10.3390/jpm11070637_

Round 1
Reviewer 1 Report
see the attached file

Author Response
On behalf of all the authors, I would like to thank you for all the information, comments and recommendation in the review. They are not only useful in this article but also in my future work as I am at the beginning of my research steps.
Grammar changes have been made for mentioned fragments as well as the whole text has been revised again.
Here are the answers to all raises questions and issues:
- Grammar changes have been made for mentioned fragments in the abstract:
- P1L23-24, grammar: “the difference … differ”
The gene expression of NRXN1 at both mRNA and protein level significantly differs and it is lower in experimental group from comparing to the expression in healthy people. The difference in gene expression of NRXN1 at both mRNA and protein levels between the first and second measurement in experimental group is also significant.
- P1L26-27, grammar: “while reducing the severity of depression”
Result demonstrates a higher expression level in the first measurement and lower expression level in the second measurement when reported depression symptom are less severe.
- P1L28-29, grammar, syntax: “… levels may play an important role in the etiopathogenesis of depressive disorders such as looking for depression biomarkers and identifying evidence relevant to personalizing treatment for depression.”. The sentence is hardly
- Results concerning expression of NRXN1 may play an important role in further researches about the etiopathogenesis of depressive disorders such as looking for depression biomarkers and identifying evidence which may be relevant to personalize treatment for depression.
- P1L30: the sentence “These expressions significantly differ from the expressions of these genes in healthy ” – linguistically incorrect and unnecessary in this part.
It was deleted.
- Grammar changes have been made for mentioned fragments in the introduction.
The backgrounds are presented adequately. Perhaps the potential role of neurexins in addiction, ADHD and Tourette syndrome would be worth mentioning in this section. Given the fact that the structural organization of pre- and post-synaptic machinery is highly complex, the interaction of neurexin with other compounds, e.g., neuroligin, synaptotagmin, scaffolding proteins, would be wise to note, as well as its role in synaptogenesis.
Information about neuropsychiatric diseases and synaptogenesis with a reference have been mentioned in the text.
Also:
- P2L75, grammar: “… ” Compared to what?
Levels of NRXN1 expression in the prefrontal cortex are changed in schizophrenia and bipolar disorder compared to healthy people. - P2L92-95, grammar, syntax: “The purpose of the study was also to investigate the role of the NRXN1 gene in the etiology and epigenetics of depression looking for depression biomarkers and identifying evidence relevant to personalizing treatment for depression.” The Authors should clarify how the results of the study link to the personalized treatment of
It may help to look for depression biomarkers and to identify evidence which may be relevant to personalize treatment for depression such as targeted therapies, especially in treatment-resistant depression.
- Material: The same rules of exclusion were also applied to the control group. All study participants were familiarized with the information about the study, have given an informed consent to participate in the research. They could withdraw it at any stage of the study without any consequences, including further treatment in the experimental group. The abbreviations for the study and the control group have been changed. The aim of double blood sampling in the experimental group was to verify if the expression of the studied gene may change when the improvement in mental state was expected and reported depression symptoms were less severe. Information about sociodemographic variables has been expanded.
This section should be supplemented with the exclusion criteria for controls. No information on whether the patients have given an informed consent to participate in the study. The abbreviations for the study and the control group should be unified (DD, EG both present in the text). The aim of double blood sampling should be clarified, current reasoning seems difficult to follow. The sociodemographic variables need to be enumerated. Furthermore:
- 3L106, repetition: “data on course of disorder…” already mentioned in the previous sentence
Data on the illness course (duration of the disease, number of hospitalizations and number of episodes) were obtained using the Composite International Diagnostic Interview (CIDI). In addition, information about sociodemographic variables
(age, sex, education, place of residence) was collected.
- P3L112, substantial error: “The aim of double blood sampling in the control group…”
The aim of double blood sampling in the experimental group was to verify if the expression of the studied gene may change (…)
- P3L113, speculation: “due to the expected improvement in mental state.”
(…) when the improvement in mental state was expected and reported depression symptoms were less severe. It was also related to the end of treatment in the hospital.
- P3L114, explanation of abbreviation missing
SSRIs (selective serotonin reuptake inhibitor)
- P3L114, missing reference: “standard antidepressant treatment”. Standard on what basis?
All the patients were treated with typical antidepressant treatment, mostly with SSRIs (selective serotonin reuptake inhibitor).
[I have changed to word “typical”. None of atypical antidepressants were used.]
- P3L114-115, repetition: “Sociodemographic variables and data on course of disorder were also ” Already mentioned twice in the text.
It was deleted.
- Methods:
The Authors described sufficiently the RNA isolation techniques and provided substantial specifications of the utilized reagents.
- P3L128, language: “determination … was determined”
Concentration of NRXN1 protein in the serum of patients and controls was determined by ELISA test, using the Human NRXN1 Elisa kit (MyBiosource, San Diego CA, USA) in accordance with the manufacturer's recommendations
- Results: P values for coefficients correlations and t-test have been added in the text, another statistical data have been checked and improved in the text. Regression information have been added to the results description. All data concerning statistical analyses and results have been checked and improved. Graphs have been added in better resolution. Characteristics of studied groups have been moved to the material section.
While the results seem sound in a certain portion, the language is highly unclear and the data has been displayed vaguely. The Authors should rewrite this section to highlight the most prominent findings. Moreover, the detailed characteristics of studied groups should be moved to the material section. No confidence intervals for the measures of spread nor for regression coefficients have been provided. P-values for statistically significant differences should be included in the text. Analyzed clinical features must be well defined and characterized (mean, median, range, standard deviation). Furthermore:
- P4L170, replace cardinal with ordinal numbers, remove the parentheses
It has been changed.
- P5L175-180, grammar: rewrite into a single statement providing the Spearman’s rank correlation coefficients and p-values in parentheses. Also clarification on the aim of given analysis should be made, since the fact two variables correlate cannot prove
It has been changed.
- P5L181-185, grammar: as above
It has been changed.
- Discussion: The aim to repeat and discuss all the results was to summarize the entire methodological part. The possibility of a more extensive discussion is limited by the amount of available literature, especially lack of the studies where expression of NRXN1 in the group of patients diagnosed with depression were conducted, however, there are reference to the results of researches with other neuropsychiatric diseases.
Certainly the least favorable part of the paper providing no analytical depth. The Authors literally reiterated the results without drawing any plausible conclusions. Scarcity of adequately selected references impedes placing the results of the study in the context of published literature.
- P8L262-264, missing reference
This sentence was changed.
- P8L269, listed publications do not support the statement
This sentence was changed.
- P8L273, listed publications do not support the statement, inference errors
This sentence was changed.
- P9L286-289, substantial error, correlation does not imply prediction nor causation
All the correlation where conducted to extend the previously presented results in t-test and to emphasize the relationship.
- P9L296-299, unfounded conclusion: “the results of the HDRS test were higher in the older group of respondents, which may indicate an increase in the severity of depressive symptoms with age, what is associated with an increasingly frequent diagnosis of depression in the elderly and decreased quality of life in this group.”
This sentence was added with a reference as a hypothesis and a probable explanation of the obtained results - it is not the final conclusion.
- P9L306, too speculative: “may lead to new methods of prevention.”
It has been changed.
- P9L312-314, inconsistency: “A significant limitation may also be the possibility of qualifying people with co-morbidity affecting the obtained results.” Weren’t the exclusion criteria applied upon inclusion?
Exclusion criteria were also applied upon inclusion but we assumed that not all
could be checked or verified during the study, especially these which were
checked using provided questionnaire or interview.

Reviewer 2 Report
The research presented in this manuscript is well designed, but the results are not well presented and explained. There are several flaws in data analysis and results.
1) Results are not well summarised in the abstract.
2) Researchers have used t-test (parametric) and for correlation - spearman rank correlation (non-parametric); how is this justified?
3) P values for correlation coefficients are missing.
4) Regression tables and results: Most important element of regression is missing (beta), and results are not well explained.
5) There is too much statistical analyses, not well articulated and explained. Therefore should be explained well and rearranged.
6) P values have not been written down properly.
7) Graphs (figures) are all blurred.
8) There are several grammatical issues.
Author Response
We would like to thank for all the information and recommendation in the review. Here are the answers to all raises questions:
- The description of the research results in the abstract has been changed.
- We decided to use Spearman Test for correlations because it is more resistant to outliers in trials than the Pearson correlation. It also shows any monotonic dependence - also non-linear and it’s often used in researches.
- P values for coefficients correlations have been added in the text.
- Regression information have been added to the results description.
- All data concerning statistical analyses and results have been checked and improved.
- P values have been checked and improved in the text.
- Graphs have been added in better resolution.
- Text has been checked for grammar mistakes.

Round 2
Reviewer 1 Report
see the attached file

Author Response
It would be advisable to provide confidence intervals with standard deviations.
In the text and tables where it is need there is information about SD.
- P5L192-195: it is still not entirely clear between what variables the analysis was performed
This sentence has been rewritten.
- P6L212: “There is no significant relationship between expression of the studied gene and sex” – contradictory to the graph and the information above.
Graph 2 describes information about age and expression of the studied gene which are also written and explained above it. This sentence is another piece of information which was added to the articles due to the whole conducted study – it hasn’t become significant.
- P6L213: “There is also no significant relationship between clinical variables and expression of the studied gene in DD group.” – data not shown.
This is the same information as mentioned above – it can be helpful for another study to verify it once again. Our goal was to shown the most important data.
- The Authors should consider adding a supplementary graph on the expressions of the
studied gene in the experimental group as in the healthy controls separately.
This was considered but was finally deleted because of too many graphs according to another review.
The referee agrees with the Authors that no other research on NRXN1 gene in depressed population was published. Indeed, the novelty is the strongest asset of the paper. However, several studies on that gene in different neuropsychiatric disorders are available. In schizophrenia and bipolar disorder the levels of neurexin are reported to be increased in contrast to the Authors’ findings in depressed population – why is that so?
What might have biased the research? What methodological and statistical differences could be observed between studies?
The Authors may also hypothesize about the impact of potential neuroligin depletion in depression on the synaptic machinery and its place in depression development.
The discussion has been expanded.
- P10L305-307: contradictory to P7L234-236
It has been changes to selected data.
- P10L312-313: “The last research aim … to assess the impact of sociodemographic
and clinical variables.” – already mentioned in the discussion.
We left only other sociodemographic variables there – age vs HDRS level.
- P10L314-317: remove or rewrite if necessary. In current form not understandable.
It has been rewritten.
Reviewer 2 Report
There are still several issues in the way results from statistical data analysis have bene presented. Conventions for symbols and abbreviations must be adhered to. Figures are now clear, but are not complete in terms of providing information on significant differences, which should be indicated above the bars. P values for table 3?
There is no need to say, table No. 5th or graph No. 6th, simple write Table 5.
Author Response
- There are still several issues in the way results from statistical data analysis have bene presented. Conventions for symbols and abbreviations must be adhered to. Figures are now clear, but are not complete in terms of providing information on significant differences, which should be indicated above the bars.
Our aim was to present all the important data in tables and make graphs the supplementary items.
- P values for table 3? P value for table 3 was written in the text (Expression of NRXN1 in the 1st measurement at the protein level correlates with expression of NRXN1 in the 2nd measurement at the mRNA level and at the protein level. Both relationships are very strong (Spearman's rank correlation coefficient = 0.89 and 0.88; p<0.05), I have been added also to the table.
- There is no need to say, table No. 5th or graph No. 6th, simple write Table 5.
This change comparing to the first version of the manuscript has been made due to another reviewer's indication .